# Nutritional Factors and Food and Nutrition Insecurity in Patients with Tuberculosis

**DOI:** 10.3390/nu17050878

**Published:** 2025-02-28

**Authors:** Yasmim Costa Mendes, Ana Larysse Lacerda Dourado, Patricia Vieira de Oliveira, Aline de Oliveira Rezende, Amanda Caroline de Souza Sales, Gabriel Pereira de Sousa, Elaíne de Araújo Pereira, Elane Luiza Costa Sousa, Maria Cecília Cruz Morais Lindoso, Roberdilson de Melo Rodrigues Júnior, Letícia Rocha Fernandes, Luciana Cabral Santana, Millena Ferreira Goiano, Luís Cláudio Nascimento da Silva, Rafiza Félix Marão Martins, Eduardo Martins de Sousa, Adrielle Zagmignan

**Affiliations:** 1Graduate Program in Biosciences Applied to Health, CEUMA University, São Luís 65075-120, MA, Brazil; larysselacerda21@gmail.com (A.L.L.D.); patriciavo94@outlook.com (P.V.d.O.); amanda_salles15@hotmail.com (A.C.d.S.S.); elane104993@ceuma.com.br (E.L.C.S.); luhcabral148@gmail.com (L.C.S.); luiscn.silva@ceuma.br (L.C.N.d.S.); eduardo.martins@ceuma.br (E.M.d.S.); 2Graduate Program in Health Sciences, Federal University of Maranhão, São Luís 65085-580, MA, Brazil; alinerezendee20@gmail.com; 3Graduate Program in Medicine, CEUMA University, São Luís 65075-120, MA, Brazil; gabrielpsousa.med@gmail.com; 4Graduate Program in Nursing, CEUMA University, São Luís 65075-120, MA, Brazil; elaine.arauj25@gmail.com (E.d.A.P.); cecilia.cruzmorais@hotmail.com (M.C.C.M.L.); roberdilsonmrjr@gmail.com (R.d.M.R.J.); leticia026312@ceuma.com.br (L.R.F.); 5Graduate Program in Biomedicine, CEUMA University, São Luís 65075-120, MA, Brazil; mgoiano130@gmail.com; 6Graduate Program in Management and Health Care, CEUMA University, São Luís 65075-120, MA, Brazil; rafiza.felix@ceuma.br

**Keywords:** *Mycobacterium tuberculosis*, malnutrition, nutritional status

## Abstract

**Background/Objectives**: Brazil has a high incidence of new tuberculosis cases influenced by socioeconomic factors. Inadequate housing, limited access to health services, and insufficient food increase vulnerability to the disease. This study aimed to identify sociodemographic, nutritional, and anthropometric factors associated with active pulmonary tuberculosis. **Methods**: This case–control study was conducted at the tuberculosis (TB) Referral Hospital in São Luís, Maranhão, Brazil, from 2022 to 2024. It included 65 patients with active pulmonary TB and 73 comparison individuals. Sociodemographic and nutritional data were collected using an adapted semi-quantitative questionnaire with 77 items, based on the validated ELSA-BRASIL questionnaire for adults, to assess the consumption of calcium, phosphorus, iron, zinc, vitamin B3, vitamin B6, vitamin C, vitamin E, and vitamin D. The Brazilian Food Insecurity Scale (EBIA) was used to analyze food and nutrition security or insecurity. Body Mass Index (BMI) was calculated based on weight and height measurements. **Results**: The TB patients were predominantly men (73.8%), of mixed ancestry (69.4%), with low education levels (73.4%), and had a mean age of 39 years. Furthermore, 26.2% of the patients were underweight, and 64% experienced some degree of food and nutrition insecurity. The variables education level, vitamin D, and BMI were associated with the outcome in the final model. **Conclusions**: The study identified a high prevalence of food insecurity and vitamin E deficiency in individuals with active tuberculosis, in addition to associating low educational levels, inadequate vitamin D intake, and overweight with a higher risk of TB. Although it has limitations, such as the cross-sectional design and sample size, the results highlight the importance of these determinants and point to the need for longitudinal research to confirm and expand the evidence.

## 1. Introduction

*Mycobacterium tuberculosis* is a facultative intracellular aerobic bacterium. It is known to trigger TB, a highly infectious and contagious disease [1]. TB transmission predominantly occurs through aerosol-like droplets spread when sneezing, coughing, or talking. Although pulmonary contamination is common, *M. tuberculosis* can affect various organs and systems. This expands its clinical manifestations and enhances challenges in diagnosing and managing the disease [2]. Despite global efforts to control TB, it remains a major public health concern. According to the latest available data, 10.6 million people worldwide were infected with TB in 2021 [3]. Additionally, 78,057 new cases were reported in Brazil in 2022. In the same year, there were 3196 new cases in Maranhão and 1427 in São Luís, the capital [4].

Brazil maintains a high incidence of TB and is among the 20 countries with the highest estimated number of new cases. This incidence is directly correlated with socioeconomic challenges [5]. Pre-existing vulnerabilities such as precarious housing, lack of access to health services, poor nutrition, unhealthy working conditions, and inadequate sanitation make individuals susceptible to developing this disease [6]. Furthermore, coinfection with the human immunodeficiency virus (HIV) can increase the risk of progression to active tuberculosis by up to 26 times [7]. Individuals in older age groups have a four-fold increased risk of developing TB [8]. Other factors, such as excessive alcohol consumption, smoking, and diabetes mellitus, are also associated with an increased incidence of tuberculosis cases [9].

As established in Article 3 of Law No. 11,346 [10] (pp. 1–2), “Food and nutritional security (FNS) consists of the realization of everyone’s right to regular and permanent access to quality food, in sufficient quantity, without compromising access to other essential needs, based on health-promoting food practices that respect cultural diversity and are environmentally, culturally, economically, and socially sustainable”. Any scenario that prevents compliance with this legislation can characterize Food and Nutrition Insecurity (FNI), making malnutrition one of the possible consequences.

Numerous studies have demonstrated the detrimental effects of malnutrition on various health conditions, including heightened susceptibility to infections, impaired healing processes, and an elevated risk of complications in inflammatory bowel diseases (IBDs) such as Crohn’s disease (CD) and ulcerative colitis (UC) [11,12,13]. These conditions are associated with deficiencies in nutrient intake, digestion, and absorption. Malnutrition is particularly concerning due to its association with worse prognoses, increased risk of infections, and higher postoperative complications [14]. Additionally, nutritional status can significantly influence nutrient–medication interactions [15] and the ability to cope with oxidative stress resulting from inadequate health conditions [16].

The bidirectional relationship between malnutrition and TB has been noted. Poor nutritional status weakens the body, making it more prone to infections. Additionally, TB can cause a decrease in food intake and malabsorption, leading to malnutrition [17,18]. Malnutrition due to micronutrient deficiencies, mainly resulting from low bioavailability and impaired absorption of essential nutrients, contributes to the imbalance of several physiological mechanisms, including those related to immune system functioning [19]. Additionally, nutritional status is closely related to disease severity, making micronutrients especially important for a positive clinical outcome [20].

Therefore, considering the scarcity of studies evaluating the determinants of the interaction between nutritional aspects and the incidence of TB, it is essential to conduct research to clarify the possible causal links between these variables. In this context, the present study aims to fill some scientific gaps by identifying sociodemographic, nutritional, and anthropometric factors associated with active pulmonary tuberculosis. It also seeks to provide an understanding of how these factors can influence not only the development of TB but also the course of the disease and the response to treatment.

## 2. Materials and Methods

### 2.1. Study Setting and Participant Population

This was a case–control study conducted with patients diagnosed with active pulmonary TB. The case group consisted of 65 patients diagnosed with active pulmonary TB, recruited from a referral hospital in São Luís, Maranhão, Brazil. To avoid the possibility of collecting individuals with Latent Tuberculosis Infection (LTBI), the comparison group was recruited from a university in São Luís, Maranhão, Brazil, totaling 73 individuals. Both groups were collected between the years 2022 and 2024.

Inclusion criteria were established for patients with active pulmonary TB: confirmed clinical and/or laboratory diagnosis of pulmonary TB, first time initiating treatment, and adulthood. The inclusion criteria were the presence of extrapulmonary TB, patients with autoimmune or immunosuppressive diseases, pregnant women, patients who did not meet the purposes of the study because they had a health status that compromised the anthropometric assessment, or who were unable to answer the questionnaires or answered incompletely.

To be eligible, participants in the control group had to be negative for TB and of the same sex and age, with a variation of up to 5 years between the two groups (case and control). The criteria for non-inclusion were those under 18 years of age, pregnant women, those with disease and/or use of immunosuppressive medications, those without diabetes, those without heart disease, and if they had hypertension, which was controlled by medication.

### 2.2. Characterization of the Clinical and Sociodemographic Profile

Sociodemographic data (gender, occupation, marital status, age group, race, education) and medical diagnosis were collected through interviews and reading of each patient’s medical records.

### 2.3. Nutritional Status Assessment

#### 2.3.1. Anthropometric Collection

Anthropometric data such as weight, height, and Body Mass Index (BMI) were determined from each study participant. Subjects were weighed without shoes and socks on a scale (OMRON)™ (Kyoto, Japan), while height was recorded using an ultrasonic measuring stadiometer (BIC)™ (Itupeva, SP, Brazil). BMI was calculated by dividing body weight (kilograms) by the square of height (meters), and for adults (20 to 59 years), according to the World Health Organization, it was classified as BMI < 18.5 kg/m^2^ = malnutrition; BMI between 18.5 and 24.9 kg/m^2^ = normal weight; BMI ≥ 25 kg/m^2^ = overweight; BMI ≥ 30 kg/m^2^ = obesity. For the elderly, it was classified as (≥60 years) BMI ≤ 22 kg/m^2^ = underweight; BMI between 22.1 and 26.9 kg/m^2^ = normal weight; BMI ≥ 27 kg/m^2^ = overweight [21,22].

#### 2.3.2. Brazilian Food Insecurity Scale (EBIA)

To assess the food and nutritional safety or insecurity of TB patients, the Brazilian Food Insecurity Scale (EBIA) was used, consisting of 14 objective questions about access to food in the family context in the past three months.

The scores and classifications were based on the presence or absence of residents under 18 years of age. Families were identified as being in a situation of food and nutrition security (FNS) with adequate access to food or in one of the three levels of food and nutrition insecurity (FNI): mild, indicating impaired access; moderate, indicating insufficiency or reduction; and severe, indicating significant restriction of food for all household members.

For households under 18 years of age, a cut-off point of 0 indicates food and nutrition security, 1 to 5 points indicate mild food and nutrition insecurity, 6 to 9 points indicate moderate food and nutrition insecurity, and 10 to 14 points indicate severe food and nutrition insecurity. For households without children under 18 years of age, FNS is also represented by 0 points, while 1 to 3 points indicate mild FNI, 4 to 5 points indicate moderate FNI, and 6 to 8 points indicate severe FNI [23].

#### 2.3.3. Food Frequency Questionnaire (FFQ)

Food intake was assessed by applying a semi-quantitative food frequency questionnaire (FFQ) with 77 items for the group with active TB and the comparison questionnaire based on the ELSA-BRASIL questionnaire already validated for adults [24]. The intake of calcium, phosphorus, iron, zinc, vitamin B3, vitamin B6, vitamin C, vitamin E, and vitamin D was estimated.

Food consumption was assessed using home measurements of each food over the last 3 months. The FFQ divided the frequency of consumption into weekly (5 to 6 times, 2 to 4 times, 1 time) and monthly (1 to 3 times, rarely, or not consumed). These frequencies were transformed into daily intake by dividing weekly intake by 7 and monthly intake by 30. The number of portions reported was multiplied by the food’s value in grams and the average number of times the interviewee reported consuming it, thus obtaining the daily consumption value [25].

A database containing the composition of each FFQ item was built in Excel to estimate the nutritional composition of the diet of each participant with active TB and of the comparison group. This database had the Household Budget Survey (POF), the Brazilian Food Composition Table (TACO), and the Food Composition Table (PHILIPPI) as reference sources.

Soon after, to verify the adequacy of intake, the recommendations of the Dietary Reference Intakes (DRIs) were used, which contain the recommended values of nutrients and energy. To classify low, adequate, or high intake, the Estimated Average Requirement (EAR) was used as a baseline, and for those nutrients that contained a Tolerable Upper Intake Level (UL) value in elevated intake, also [26,27,28].

Consumption was classified as follows: low when the values were less than 20% of the reference value of the EAR; adequate when the values were within 20% above or below the reference value of the EAR; high when the values exceeded the reference value of the EAR by 20%; and elevated when the values exceeded the UL recommendation.

### 2.4. Statistical Analysis

The analyses were performed using the STATA 15.0 statistical program. Univariate and bivariate descriptive analyses were conducted for all study variables, calculating absolute and relative frequencies. The chi-square and Fisher’s exact tests were used to verify the association between the independent variables and the outcome.

To evaluate the association between the exploratory variables and the outcome (presence of TB), we used multivariate hierarchical logistic regression analysis, adjusting the odds ratio (OR) and the respective 95% confidence intervals (95% CI).

The multivariate analysis was performed using a hierarchical approach, composed of blocks of variables at distal levels (socioeconomic and demographic), intermediate level (nutrition-related aspects), and proximal level (BMI). At all levels, only those variables that presented a descriptive level of *p* < 0.10 remained in the model after adjustment for the variables of the previous levels. In the final model, a significance level of 0.05 was adopted.

### 2.5. Ethical Aspects

The study was carried out in accordance with the norms that regulate research on human beings contained in Resolution No. 466/12 of the National Health Council and the Declaration of Helsinki II (2000) and was approved by the Research Ethics Committee (CEP) of CEUMA University (No. 5.541,104). All study participants signed the Informed Consent Form (ICF) before data collection.

## 3. Results

A total of 138 individuals were included in the study, of whom 65 corresponded to those with active TB and 73 to individuals in the comparison group, with a mean age of 39 (SD = 14.9) and 34 (SD = 11.8) years, respectively. In the group with active TB, 73.8% of the individuals were men. This characteristic was similar in the comparison group, where males predominated with 61.6%.

In relation to the other sociodemographic characteristics, there was a higher percentage of people who self-declared themselves as ‘mixed ancestry’ in the entire sample analyzed (59.2%), as well as in each group. Regarding education, the smallest group with active TB had higher education (8.2%), contrasting with the comparison group in which 78.1% had a high level of education.

This information forms a pattern since 92.2% of the individuals with active TB lived on ≤3 minimum wages, whereas almost 60% of the comparison group had more than 3 wages as income. Race (*p* < 0.002), education level (*p* < 0.000), occupation (*p* < 0.000), household income (*p* < 0.000), calcium (*p* < 0.001), vitamin D (*p* < 0.000), zinc (*p* < 0.043), and BMI (*p* < 0.000) were significantly different in the active TB group compared to the comparison group, indicating an association with the presence of the disease (Table 1).

The variables education level, vitamin D, and BMI remained associated with the outcome in the final model. People who only had completed primary education were 93 times more likely to develop TB than those who had completed higher education. A high consumption of vitamin D is associated with a 7.13 times higher risk of developing TB compared to those who do not consume vitamin D adequately, whereas those who were overweight/obese were 2.14 times more likely than those with normal weight (Table 2).

Regarding the anthropometric indicator, the group with active TB had a mean BMI of 20.91 (SD = 3.808), and the comparison group had a mean of 25.75 (SD = 4.212). Almost 60% of the individuals with active TB were eutrophic, although 19% were on the threshold of thinness (BMI < 19), and 26.2% were already underweight. On the other hand, 50.7% of the individuals in the comparison group were overweight or obese, and 47.9% were eutrophic, with only 1.4% being underweight (Figure 1).

Regarding food and nutrition security (FNS), 64% of the patients had some degree of food and nutrition insecurity, while 49.3% of individuals in the comparison group experienced food and nutrition insecurity (Figure 2).

The food consumption data were classified as low, adequate, high, or elevated according to the EAR and UL values and presented in Figure 3. Based on the results, high vitamin E deficiency was found in the active TB group and in the vitamin E and vitamin D comparison group, both in men and women, in the various age groups studied, with percentages above 70% in both groups.

## 4. Discussion

Similar results were found in the study by Slash et al. [29], where 75% of TB patients and 60% of healthy individuals were male. The higher incidence of TB among individuals aged 34 to 39 years may be related to the peak of professional activity, increasing exposure to risk environments [30]. Men are particularly vulnerable due to behaviors such as smoking and alcohol consumption and seek fewer preventive health services [31,32]. The study by Mesquita et al. [33] on new TB cases on Marajó Island showed that individuals of mixed ancestry were predominant (82%), corroborating the present study. In the study on the association of socioeconomic factors and the incidence of pulmonary TB [34], a high percentage of TB patients lived on less than two minimum wages and had only basic education.

These findings are supported by Rafiq et al. [35], who state that socioeconomic characteristics directly influence the health status of the population, thus affecting the presence and progression of the disease. A limited educational background restricts access to health information and medical services, resulting in delayed diagnosis and inadequate treatment [36]. In a case–control study conducted in China, participants had a median BMI of 19.06 kg/m^2^, impairing the prognosis of TB due to the weakening of the body’s resistance to M. tuberculosis infection [37]. In addition, Ma et al. [38] reported that patients with catabolic diseases such as TB may be malnourished even if they have a normal BMI, with the measurements found in the present study also being a possible risk factor.

In accordance with these results, in the study by Oswal et al. [39], it was documented that increased adiposity can contribute to the activation of immune cells in the lungs, reducing the bacterial load during TB infections. However, while overweight/obesity may offer some protective effects against TB, it is also associated with other health risks, such as diabetes, which can intensify the severity of TB, hence the need for further studies to explore how different dietary patterns may affect TB, especially in people with diabetes and TB [40].

Ayiraveetil et al. [41] assessed food insecurity at home among newly diagnosed TB patients and noted that about one-third of their sample had some degree of food and nutrition insecurity (FNI). They concluded that this insecurity can be both a cause and a consequence of TB. Few studies on FNI are present in the literature. However, a review by Ojo et al. [42] documented that families with FNI often forgo quality food or experience significant reductions in food quantity.

One study reaffirms the strong correlation between FNI and TB, especially in individuals with low BMI [18]. FNI, common in malnutrition contexts, exacerbates vulnerability by restricting access to adequate nutrition, weakening the immune system, and increasing susceptibility to TB bacillus infection and disease progression [43]. In addition, individuals with FNI often live in poor conditions, including overcrowded and unsanitary environments, which facilitates the transmission of the disease [44]. Socioeconomic factors such as poverty, limited access to nutritious food, and inadequate basic sanitation contribute to the relationship between FNI and TB [45].

Malnutrition is a widely recognized risk factor for the development and/or poor prognosis of TB. Thus, several studies have sought clarification on the relationship between micronutrients and this disease since the prevalence of TB in individuals with micronutrient deficiencies is an area of growing interest in public health due to the crucial role of these vitamins and minerals in the immune response [46]. Among the studies evaluating the relationship between micronutrients and TB, those most frequently included are vitamin A D, calcium, and zinc (also evaluated in this study, except for vitamin A), as there is evidence that these are more associated with the construction of immunity [47]. However, other micronutrients such as iron, vitamin C, vitamin B3, vitamin B6, and vitamin E have recently been studied [48,49].

Like iron, scientific evidence demonstrates that iron plays an essential role in immunity by regulating fundamental functions of cells involved in the innate and adaptive immune response [50]. This micronutrient influences macrophage polarization and activity, neutrophil recruitment and function, NK cell activation, and T and B lymphocyte differentiation and is, therefore, critical in defense against pathogens. Dysregulation of iron homeostasis can compromise the functionality of the immune system, favoring the development of neoplasms, contributing to diseases associated with aging, and increasing susceptibility to infections such as TB [51].

Similarly, vitamin E, a fat-soluble antioxidant, protects cell membranes from oxidative stress and reduces oxidative damage in lung tissue by regulating the production of free radicals, preserving the integrity of immune cells [52,53]. In addition, vitamin E modulates the immune response by inhibiting the proliferation of inflammatory cells and increasing lymphocyte activity and the production of cytokines that strengthen immunity. In this way, it plays an essential role in protecting and regulating the immune system [54].

Significant data on vitamin D and calcium were found in studies by Jaimni et al. [46] and Cioboata et al. [55]. Vitamin D levels were significantly lower in the active group than in the controls (*p* < 0.004), and low calcium levels were associated with increased pulmonary cavitations in more severe stages of TB (*p* = 0.049). It is evident that vitamin D deficiency significantly increases the risk of TB due to the ineffectiveness of the immune system in containing infections [56].

The available information on the role of micronutrients in tuberculosis is heterogeneous and sometimes conflicting, making it difficult to clearly understand their influence on susceptibility and disease progression. These inconsistencies may be related to methodological differences between studies, such as variations in the populations studied, diagnostic criteria, nutritional assessment methods, and supplementation levels used [57].

These findings highlight the need for further studies to provide a solid basis for effective nutritional interventions for tuberculosis control, in addition to interventions to correct micronutrient deficiencies, as well as support for NSF as part of a comprehensive TB control strategy [58]. In addition, they reinforce the need for multidisciplinary approaches and social actions, such as education and awareness programs about symptoms, forms of transmission, and the importance of treatment, reducing the stigma of the disease; programs that offer nutritional and social support, with financial assistance to patients undergoing treatment, aiming to reduce treatment abandonment due to socioeconomic difficulties.

This study contributes to the initial understanding of the complex relationship between the nutritional and socioeconomic aspects of tuberculosis. It highlights the need to strengthen existing public policies, such as national micronutrient supplementation programs, that are effective in addressing this global health challenge.

While this study provides valuable insights into the complex interplay between socioeconomic, nutritional, and anthropometric factors and active pulmonary tuberculosis, it is important to recognize certain limitations. First, the cross-sectional design limits the establishment of causal relationships between the identified factors and TB. Longitudinal studies are needed to investigate these associations further. Second, the recruitment of participants from a single referral hospital and university may not fully represent the general population. However, this study provides a valuable snapshot of the situation within these specific scenarios. Third, the use of self-reported data, such as dietary intake, may be subject to recall bias. However, this limitation is inherent in many observational studies. Finally, the relatively small sample size may limit the generalizability of the findings to other populations. Larger and more diverse studies are needed to confirm these findings in other settings.

## 5. Conclusions

This study investigated the association between sociodemographic, nutritional, and anthropometric factors and active pulmonary tuberculosis in São Luís, Maranhão, Brazil. Our findings revealed a high prevalence of food and nutrition insecurity and vitamin E deficiency among individuals with active TB. Furthermore, significant associations between lower educational levels, vitamin D, and overweight/obesity with an increased risk of active TB were observed.

These findings underscore the crucial role of socioeconomic factors and nutritional status in the development and progression of TB. Addressing these underlying issues, such as improving access to education and healthcare, promoting food security, and implementing nutritional interventions, is vital for effective TB control strategies.

## Figures and Tables

**Figure 1 nutrients-17-00878-f001:**
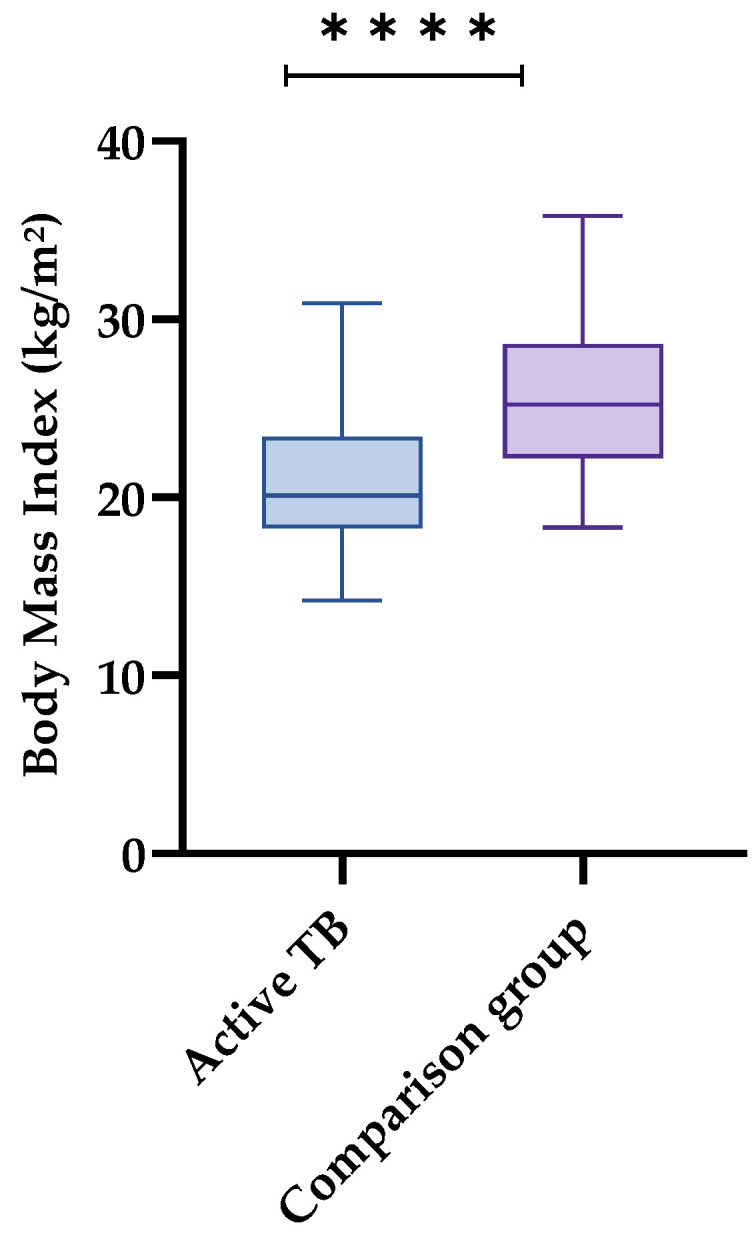
Body Mass Index (BMI) of patients with active pulmonary tuberculosis and comparison group. São Luís, MA, 2022–2024. Legend: *t*-test, **** *p* < 0.0001 (statistical significance).

**Figure 2 nutrients-17-00878-f002:**
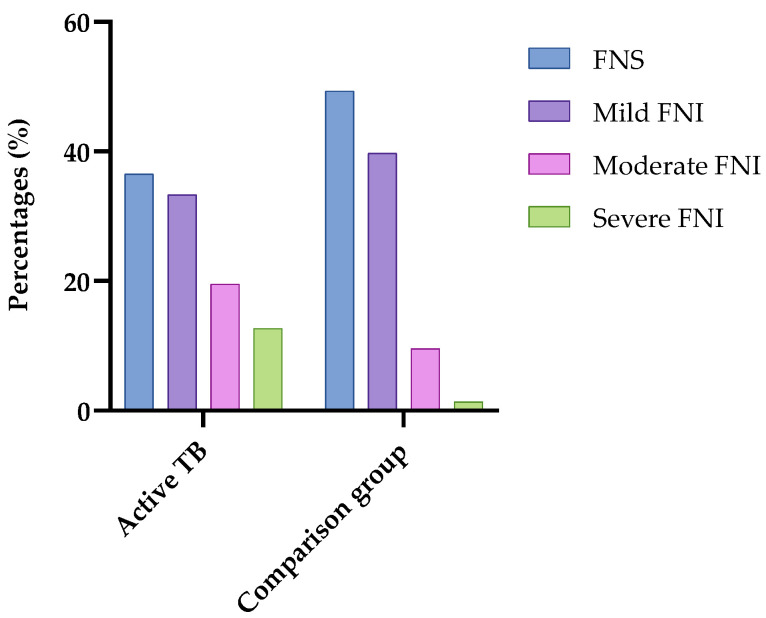
Food and nutrition safety and insecurity of patients with active pulmonary tuberculosis and comparison group. São Luís, MA, 2022–2024. Legend: FNS: food and nutrition security; FNI: food and nutrition insecurity.

**Figure 3 nutrients-17-00878-f003:**
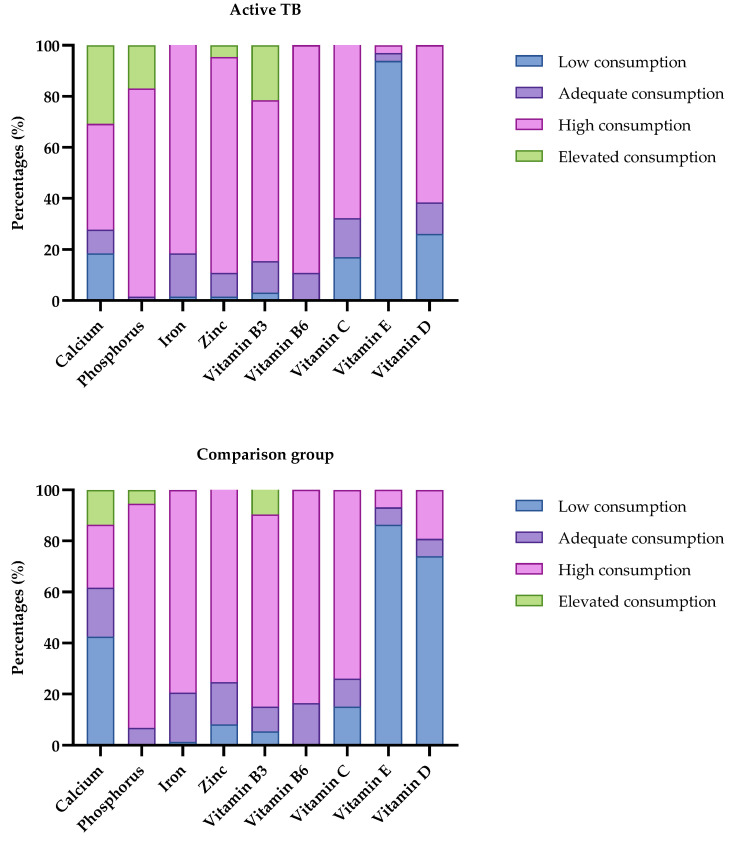
Classification of daily micronutrient intake (vitamin D, vitamin E, and calcium) of the active and comparative TB group. São Luís, MA, 2022–2024. Values are represented as a percentage (%). Legend: low consumption: 20% < EAR; adequate: ±20% EAR; high consumption: 20% > EAR; elevated consumption: >UL.

**Table 1 nutrients-17-00878-t001:** Descriptive analysis and univariate association between factors correlated with active pulmonary tuberculosis. São Luís, MA, 2022–2024.

Variables	Total		Active		Control		*p*-Value
	fa	f%	fa	f%	fa	f%	
Distal Level: Sociodemographic							
Sex							0.127
Male	93	67.3	48	73.8	45	61.6	
Female	45	32.6	17	26.2	28	38.4	
Race							0.002 *
Black	32	23.7	16	25.8	16	21.9	
Mixed ancestry	80	59.2	43	69.4	37	50.7	
White	23	17.0	3	4.8	20	27.4	
Education level							0.000 *
Higher education	62	46.2	5	8.2	57	78.1	
Middle school	44	32.8	31	50.8	13	17.8	
Elementary School	28	20.9	25	22.6	3	4.1	
Occupation							0.000 *
Employee	113	83.7	43	69.4	70	95.9	
Retired	8	5.9	5	8.1	3	4.1	
Unemployed	14	10.3	14	22.6	0	0.0	
Household income							0.000 *
>3 Minimum Wages	47	37.9	4	7.8	43	58.9	
≤3 Minimum Wages	77	62.1	47	92.2	30	41.1	
Intermediate Level: Nutritional aspects							
EBIA							0.115
Mon. Feed	59	43.0	23	35.9	36	49.3	
Insg. Food (Mild, Moderate or Severe)	78	56.9	41	64.1	37	50.7	
Calcium							0.001 *
Low consumption	43	31.16	12	18.46	31	42.47	
Adequate consumption	20	14.49	6	9.23	14	19.18	
High consumption	45	32.61	27	47.54	18	24.66	
Elevated consumption	30	21.74	20	30.77	10	13.70	
Vitamin D							0.000 *
Low consumption	71	51.45	17	26.15	54	73.97	
Adequate consumption	13	9.42	8	12.31	5	6.85	
High consumption	54	39.13	40	61.54	14	19.18	
Vitamin E							0.342
Low consumption	124	89.86	61	93.85	63	86.3	
Adequate consumption	7	5.07	2	3.08	5	6.85	
High consumption	7	5.07	2	3.08	5	6.85	
Zinc							
Low consumption	7	5.07	1	14.29	6	8.22	0.043 *
Adequate consumption	18	16.44	6	9.23	12	16.44	
High consumption	110	75.34	55	84.62	55	75.34	
Elevated consumption	3	4.62	3	4.62	0	0	
Iron							
Low consumption	2	1.45	1	1.54	1	1.37	0.941
Adequate consumption	25	18.12	11	16.92	14	19.18	
High consumption	111	80.43	53	81.54	58	79.45	
Phosphorus							
Adequate consumption	6	4.35	1	1.54	5	6.85	
High consumption	117	84.78	53	81.54	64	87.67	
Elevated consumption	15	10.87	11	16.92	4	5.48	
Proximal Level: Anthropometric aspect							
BMI							0.000 *
Eutrophy	72	52.1	37	56.9	35	48.0	
Low weight	18	13.0	17	26.1	1	1.4	
Overweight and obesity	48	34.7	11	16.9	37	50.7	

Legend: Chi-square test, * *p* ≤ 0.1 (statistical significance); fa: absolute frequency; f%: percentage frequency.

**Table 2 nutrients-17-00878-t002:** Initial and final linear regression model of the association between exposure variables and active pulmonary tuberculosis. São Luís, MA, 2024.

	Initial Model		Final Model	
	Baseline OR (95% CI)	*p*-Value	Final OR (95% CI)	*p*-Value
Distal Level: Sociodemographic				
Race				
White	0.34 (0.34–3.40)	0.358 *		
Education level				
Middle school	30.76 (5.52–171.44)	0.000 *	34.92 (7.29–167.02)	0.07
Elementary School	83.53 (9.46–737.14)	0.000 *	119.98 (17.19–836.97)	0.00 *
Occupation				
Retired	0.65 (0.084–5.08)	0.684		
Income				
≥Minimum Wages	2.20 (0.394–12.255)	0.369		
Intermediate Level: Nutritional aspects				
EBIA				
Inseg. Food (Mild. Moderate or Severe)	1.99 (0.639–6.205)	0.235		
Calcium				
Adequate consumption	0.62 (0.17–2.29)	0.470		
High consumption	0.31 (0.05–1.83)	0.195		
Elevated consumption	0.17 (0.02–1.49)	0.110		
Vitamin D				
Adequate consumption	10.56 (1.80–61.93)	0.01 *	7.13 (0.83–61.21)	0.07
High consumption	26.45 (4.28–163.52)	0.00 *	16.85 (3.72–76.20)	0.00 *
Proximal Level: Anthropometric aspect				
BMI				
Low weight	5.89 (0.57–60.73)	0.136	4.21 (0.42–41.71)	0.16
Overweight/obesity	0.08 (0.18–0.41)	0.002 *	0.065 (0.012–0.33)	0.00 *

Legend: Chi-square test, * *p* ≤ 0.05 (statistical significance); OR: odds ratio; CI: confidence interval.

## Data Availability

All data are contained in this article. For additional information, contact the corresponding authors.

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
