# Peer review of "Nutritional Factors and Food and Nutrition Insecurity in Patients with Tuberculosis"

_nutrients, 2025, doi:10.3390/nu17050878_

Round 1
Reviewer 1 Report
Comments and Suggestions for Authors
The study provides insights into the relationship between socioeconomic factors, nutrition, and the incidence of active pulmonary tuberculosis (TB) in Brazil.
The paper includes only 65 patients with TB and 73 comparison individuals. While this may provide preliminary insights, the relatively small sample size limits the generalizability of the findings. A larger cohort could yield more robust results and better represent the population.
The manuscript is conducted at a TB Referral Hospital, which may introduce selection bias. Patients seeking treatment at this facility might have more severe cases or different socioeconomic backgrounds compared to those who do not seek care, skewing the results.
The use of a semi-quantitative questionnaire based on the ELSA-BRASIL questionnaire raises concerns about the reliability and validity of the data. Self-reported dietary intake can be subject to recall bias and inaccuracies, affecting the quality of the nutritional assessment.
The study identifies correlations between TB and various socioeconomic and nutritional factors but does not establish causality. More rigorous longitudinal studies are needed to determine whether these factors contribute to the development of TB or if they are simply associated with it.
While the study assesses vitamin D and calcium deficiencies, it does not consider other essential nutrients that could impact immune function and overall health, such as protein and micronutrients. A more comprehensive nutritional assessment would provide a clearer picture of the patients' dietary deficiencies.
Although the study highlights low education levels and food insecurity, it would benefit from a more in-depth analysis of other socioeconomic determinants, such as income level, employment status, and housing conditions, which could further elucidate the complex interplay between these factors and TB incidence.
While the conclusion calls for public policies to improve nutrition and social conditions, it lacks specific recommendations or strategies for implementation. A more detailed discussion on how to address these issues effectively would strengthen the study's impact.
Author Response
We would like to express our sincere gratitude to Reviewer 1 for their insightful comments and suggestions, which have significantly improved the quality of our manuscript.
- Reviewer Comment: The study includes only 65 patients with TB and 73 comparison individuals. While this may provide preliminary insights, the relatively small sample size limits the generalizability of the findings. A larger cohort could yield more robust results and better represent the population.
Response: We acknowledge that the small sample size and cross-sectional design are limitations of the study. However, the design does not prevent us from establishing associations through a hierarchical analysis, which adds some reliability to our findings. We have included this limitation in the discussion section. We appreciate the reviewer’s insight on this matter.
- Reviewer Comment: The manuscript is conducted at a TB Referral Hospital, which may introduce selection bias. Patients seeking treatment at this facility might have more severe cases or different socioeconomic backgrounds compared to those who do not seek care, skewing the results.
Response: In São Luís, where the study was conducted, almost all TB cases are referred to this hospital regardless of disease severity. This reduces the potential for selection bias. We have clarified this in the manuscript. Thank you for highlighting this important consideration.
- Reviewer Comment: The use of a semi-quantitative questionnaire based on the ELSA-BRASIL questionnaire raises concerns about the reliability and validity of the data. Self-reported dietary intake can be subject to recall bias and inaccuracies, affecting the quality of the nutritional assessment.
Response: Recall bias is an inherent limitation in studies using self-reported dietary intake, such as those using the ELSA-BRASIL questionnaire. Despite this, the questionnaire is widely used in epidemiological and dietary consumption studies. We have included this limitation in the discussion section. We appreciate the reviewer’s attention to detail.
- Reviewer Comment: The study identifies correlations between TB and various socioeconomic and nutritional factors but does not establish causality. More rigorous longitudinal studies are needed to determine whether these factors contribute to the development of TB or if they are simply associated with it.
Response: This study is cross-sectional, and we use the term "association" accordingly. While this is a limitation, we conducted a hierarchical analysis to reduce biases. We have emphasized this in the manuscript. Thank you for this valuable feedback.
- Reviewer Comment: While the study assesses vitamin D and calcium deficiencies, it does not consider other essential nutrients that could impact immune function and overall health, such as protein and micronutrients. A more comprehensive nutritional assessment would provide a clearer picture of the patients' dietary deficiencies.
Response: Our objective was to analyze micronutrients. In this version, we have included additional micronutrients collected in the study, such as calcium and zinc, which are important for immunity. However, the lack of macronutrient (protein) analysis is a limitation, which we have included in the discussion. We appreciate the reviewer’s suggestion for a more comprehensive assessment.
- Reviewer Comment: Although the study highlights low education levels and food insecurity, it would benefit from a more in-depth analysis of other socioeconomic determinants, such as income level, employment status, and housing conditions, which could further elucidate the complex interplay between these factors and TB incidence.
Response: We evaluated income level (household income) and employment status (occupation) in the study, as described in the results (Tables 1 and 2). However, housing conditions were not a focus of this study and are a limitation. We have included this in the discussion. Thank you for pointing out this area for improvement.
- Reviewer Comment: While the conclusion calls for public policies to improve nutrition and social conditions, it lacks specific recommendations or strategies for implementation. A more detailed discussion on how to address these issues effectively would strengthen the study's impact.
Response: We have included specific recommendations and strategies for public policies in the discussion section. We appreciate the reviewer’s input on enhancing the study’s impact.
Reviewer 2 Report
Comments and Suggestions for Authors
This manuscript aims identify sociodemographic, nutritional, and anthropometric factors associated with active pulmonary tuberculosis. It also seeks to provide an understanding of how these factors can influence not only the development of TB but also the course of the disease and the response to treatment.
Please consider the following suggestions:
Please reconsider the affiliations in order to comply to the journals’ requirements.
Abstract - Please insert the abbreviations in brackets to signal their use from then on (e.g. TB)
Please consider the limitations of the study and rephrase the conclusions from the Abstract section.
I believe that information regarding other diseases that can be related to malnutrition should be mentioned in the Introduction section as well as other factors that lead to an increase in cases of TB should be mentioned.
You state that “This case-control study was conducted at a TB Referral Hospital 21 São Luís, Maranhão, Brazil, from January 2021 to July 2022.”, and then that “The case group consisted of 65 patients with a positive diagnosis for active pulmonary TB, recruited from a referral hospital in São Luís, Maranhão, Brazil, from August 2022 to July 2023. The control group consisted of 73 individuals recruited from a university in São Luís, Maranhão, Brazil, from January 2023 to February 2024.” Which is it?
Please explain why you chose the case group and the control group from different institutions and different period of time for your study.
Please detail in the Discussions section how the deficit of several nutrients relates to the TB diagnosis. Do other nutrients (except the ones you mentioned) play a role in the immunity status? Which are they and why you only referred to some of them?
Comments on the Quality of English LanguagePlease read the manuscript and correct any errors in the English language.
Author Response
We would like to extend our heartfelt thanks to Reviewer 2 for their valuable comments and suggestions, which have greatly enhanced the clarity and depth of our manuscript.
- Reviewer Comment: Please reconsider the affiliations to comply with the journal’s requirements.
Response: We have updated the affiliations to comply with the journal's guidelines. Thank you for this important reminder.
- Reviewer Comment: Please insert the abbreviations in brackets to signal their use from then on (e.g., TB).
Response: We have inserted the requested abbreviations in the abstract. We appreciate the reviewer’s attention to detail.
- Reviewer Comment: Please consider the limitations of the study and rephrase the conclusions from the Abstract section.
Response: We have included the study's limitations and rephrased the conclusions in the abstract. Thank you for this valuable suggestion.
- Reviewer Comment: I believe that information regarding other diseases that can be related to malnutrition should be mentioned in the Introduction section as well as other factors that lead to an increase in cases of TB.
Response: We have included information on other diseases related to malnutrition and additional factors contributing to TB incidence in the introduction. We appreciate the reviewer’s comprehensive feedback.
- Reviewer Comment: You state that “This case-control study was conducted at a TB Referral Hospital in São Luís, Maranhão, Brazil, from January 2021 to July 2022.”, and then that “The case group consisted of 65 patients with a positive diagnosis for active pulmonary TB, recruited from a referral hospital in São Luís, Maranhão, Brazil, from August 2022 to July 2023. The control group consisted of 73 individuals recruited from a university in São Luís, Maranhão, Brazil, from January 2023 to February 2024.” Which is it?
Response: The recruitment was conducted between 2022 and 2024. We have corrected this information in the text. Thank you for pointing out this inconsistency.
- Reviewer Comment: Please explain why you chose the case group and the control group from different institutions and different periods of time for your study.
Response: To reduce the possibility of recruiting individuals with Latent Tuberculosis Infection (LTBI), the comparison group was recruited from a university, while the active TB group was recruited from a referral hospital that receives almost all notified cases of active and latent TB in the state. We have clarified this in the methodology and abstract. We appreciate the reviewer’s insight on this matter.
- Reviewer Comment: Please detail in the Discussions section how the deficit of several nutrients relates to the TB diagnosis. Do other nutrients (except the ones you mentioned) play a role in the immunity status? Which are they and why you only referred to some of them?
Response: We have included detailed information on how nutrient deficits relate to TB diagnosis and discussed other relevant nutrients in the discussion section. Thank you for this valuable suggestion.
Round 2
Reviewer 1 Report
Comments and Suggestions for Authors
Considering the improvements made in the manuscript
Reviewer 2 Report
Comments and Suggestions for Authors
The manuscript was improved.